# Nanocelluloses and Their Applications in Conservation and Restoration of Historical Documents

**DOI:** 10.3390/polym16091227

**Published:** 2024-04-27

**Authors:** Ana P. S. Marques, Ricardo O. Almeida, Luís F. R. Pereira, Maria Graça V. S. Carvalho, José A. F. Gamelas

**Affiliations:** 1Chemical Engineering and Renewable Resources for Sustainability, Department of Chemical Engineering, University of Coimbra, Polo II, Rua Sílvio Lima, 3030-790 Coimbra, Portugal; amarques@qui.uc.pt (A.P.S.M.); ralmeida@eq.uc.pt (R.O.A.); mgc@eq.uc.pt (M.G.V.S.C.); 2Techn&Art, Polytechnic Institute of Tomar, Quinta do Contador, Estrada da Serra, 2300-313 Tomar, Portugal; fpereira@ipt.pt

**Keywords:** nanocelluloses, historical documents, paper conservation, paper restoration, iron gall ink

## Abstract

Nanocelluloses have gained significant attention in recent years due to their singular properties (good biocompatibility, high optical transparency and mechanical strength, large specific surface area, and good film-forming ability) and wide-ranging applications (paper, food packaging, textiles, electronics, and biomedical). This article is a comprehensive review of the applications of nanocelluloses (cellulose nanocrystals, cellulose nanofibrils, and bacterial nanocellulose) in the conservation and restoration of historical paper documents, including their preparation methods and main properties. The novelty lies in the information collected about nanocelluloses as renewable, environmentally friendly, and sustainable materials in the field of cultural heritage preservation as an alternative to conventional methods. Several studies have demonstrated that nanocelluloses, with or without other particles, may impart to the paper documents excellent optical and mechanical properties, very good stability against temperature and humidity aging, higher antibacterial and antifungal activity, high protection from UV light, and may be applied without requiring additional adhesive.

## 1. Introduction

Global solutions to replace the use of petroleum-based chemicals and products are growing to solve environmental and ecological problems. Nanocelluloses are promising green materials that can be used in many fields, such as packaging, paper, textiles, paint, aerospace, photonics, and excipients [1,2,3]. The main characteristics of nanocelluloses are their renewability, abundance, biocompatibility, chemical inertness, high Young’s modulus, low density, high tensile strength, large specific surface area, dimensional stability, low coefficient of thermal expansion, and tunable surface chemistry [4,5,6,7]. Nanocelluloses are fibrous materials with widths in the nanometer range [8], mainly obtained from cellulose, the most abundant polymeric raw material on Earth [9].

The preservation of historical and cultural heritage, including artifacts, artworks, manuscripts, books, sculptures, paintings, historical buildings, and archaeological finds, which are susceptible to degradation over time, is very important for society. A large part is made of cellulose-based materials (wood, paper, archaeological fabrics, and painting canvases) and requires intervention due to (i) chemical acidification processes caused by primers, paints, and glues that decrease the mechanical properties, (ii) the absorption of acid gases present in the atmosphere like sulfur dioxide and nitric oxides that increase the brittleness, (iii) biological attacks by microorganisms such as bacteria and fungi that also induce degradation [10,11]. 

Nanocelluloses are one of the most recent novelties in the field of conservation and restoration of cultural heritage, considering that many types of artistic substrates consist mainly of cellulose [12,13]. The use of paper for recording information is still very important, but the deterioration of this substrate can be a serious problem for world libraries, archives, and museums [14,15,16]. Some illustrative examples are the acidic paper manufactured between the middle of the nineteenth and twentieth centuries [16,17] and paper documents written with iron gall inks that have been used extensively from medieval times until the nineteenth century [18,19].

## 2. Cellulose and Its Isolation

Cellulose is considered the most abundant renewable source on Earth, with an estimated annual production of 1 × 10^10^ to 1 × 10^11^ tons. However, only a small amount, around 6 × 10^9^ tons, is processed by the industry of paper, textiles, chemicals, and other materials [20]. The sources of cellulose are innumerous, namely wood, herbaceous plants, grass, agricultural crops and their by-products, animals, algae, tunicates, fungi, bacteria, and waste paper [21,22]. Cellulose is a white biomacromolecule with no odor and no taste. It is insoluble in water and most organic solvents, and it has a density of around 1.5 g/cm^3^, independent of the source [23].

Chemically, cellulose is a linear homopolysaccharide produced from glucose monomers linked by β-1,4-glycosidic bonds. Depending on the source, the number of anhydroglucopyranose units can reach values up to 10,000. Cellulose chains are held together by hydrogen bonds and van der Waals forces, presenting highly ordered (crystalline) regions and disordered (amorphous) regions. Hydrogen bonds arise between oxygen and hydroxyl groups positioned within the same cellulose molecule (intramolecular) and between neighboring cellulose chains (intermolecular) [24]. Intermolecular hydrogen bonds are responsible for the physical properties of cellulose, such as good strength and flexibility. Crystalline regions are more resistant to chemical, mechanical, and enzymatic treatments, have a higher resistance to degradation, and have a lower solubility in water and other solvents [6,25]. Amorphous regions are more likely to react with other molecular groups [26,27,28].

Native cellulose has a well-organized and aligned structure composed of cellulosic fibers, macrofibrils, microfibrils, nanofibrils, and elementary fibrils (Figure 1). The arrangement into distinct layers does not exist in regenerated cellulose because they are randomly positioned in the structure. Cellulosic fibers can be laterally disintegrated by mechanical processes to form cellulose nanofibrils or can be cleaved transversally at the less ordered regions to form cellulose nanocrystals [29].

The isolation of cellulose from plant-based sources involves separating cellulose fibers from non-cellulosic components. The non-cellulosic components are essentially lignin and hemicelluloses that can typically be present in cell walls. Lignin is a complex aromatic polymer that surrounds cellulose and acts as a barrier, conferring recalcitrance to the matrix. Hemicelluloses are branched carbohydrate polymers found alongside cellulose. The plant source material is prepared by removing bark or leaves and reducing its size, making it easier to handle. Then, the material is dried to decrease the moisture content, since a high moisture content can lead to microbial growth and degradation of cellulose. The prepared source material is subjected to pulping through mechanical or chemical methods, which consist of breaking down the bonds between the cellulose fibers and the non-cellulosic components. Mechanical methods rely on physical forces to separate the cellulose fibers, and chemical methods include the use of chemicals to dissolve or degrade the non-cellulosic components. The cellulose fibers are then washed to remove impurities, residual chemicals, or other by-products and dried, making them suitable for use. Methods of cellulose isolation depend on the source, desired purity, properties, and applications.

## 3. Nanocelluloses

Nanocelluloses are defined as cellulose fibers with at least one dimension at the nanoscale (1–100 nm). The versatility of nanocelluloses allows them to have a wide range of applications. Nanocelluloses properties are generally better compared to those of cellulose due to the reduced size that leads to structure and behavior changes [31,32].

Three main types of nanocelluloses can be distinguished: (i) cellulose nanocrystals (CNCs) or nanocrystalline cellulose or cellulose whiskers; (ii) cellulose nanofibrils (CNFs) or nanofibrillated cellulose or cellulose nanofibers or microfibrillated cellulose; (iii) bacterial nanocellulose (BNC) or microbial cellulose [33]. CNCs are composed essentially of crystalline regions (crystallinity index of 54–88%) possessing a well-ordered structure; CNFs consist of both crystalline and amorphous regions (crystallinity index < 50%) having a more random structure; and BNCs are composed of almost pure cellulose with minimal impurities or amorphous regions (crystallinity index > 88%) possessing an exceptionally well-organized structure [34,35]. The choice between each type of nanocellulose depends on the unique characteristics that can be used in specific applications. CNCs and CNFs are produced through plant-based cellulose sources requiring the isolation of cellulose from lignin and hemicelluloses, unlike BNC, which is obtained through bacteria species from sugars and does not have unwanted polymers [36,37,38]. The preparation processes of the three types of nanocelluloses are different because CNCs and CNFs are formed from the largest to smallest dimensions, called the top–down process, and BNCs are created from the smallest to largest dimensions, designed as the bottom–up process [39]. Table 1 presents the main differences between the three types of nanocelluloses.

### 3.1. Cellulose Nanocrystals

CNC production involves breaking down the fibers, resulting in nanocrystals. The amorphous regions of cellulose are removed, contributing to a high degree of crystallinity, excellent mechanical properties, a high surface area, and good thermal stability [43,44,45]. CNCs consist of cylindrical, elongated, and inflexible particles with a rod-like structure and dimensions of 50–500 nm in length and 3–50 nm in diameter. 

#### 3.1.1. Acid Hydrolysis

Acid hydrolysis is the most common method used to produce CNCs from cellulose fibers. Cellulose fibers are mixed with an aqueous acid solution at an appropriate concentration and temperature, which breaks down the amorphous regions by cleaving the glycosidic bonds between the glucose units, leading to the formation of nanocrystals. The acid hydrolysis is stopped by neutralizing the acid, typically by adding water or a basic solution (e.g., sodium hydroxide), resulting in a suspension that is washed by dialysis against deionized water to remove any remaining acid and neutralize salts. Nanocrystals are separated from the suspension through centrifugation or filtration and dried (freeze-drying or spray-drying). A mechanical treatment (sonication) is usually applied to prevent agglomeration and obtain a homogeneous dispersion [46].

The glycosidic bonds of the amorphous regions of cellulose fibers are easily hydrolyzed by acid, while the crystalline regions are preserved since the acid has great difficulty penetrating the well-ordered regions [47]. Various strong acids can be used successfully in acid hydrolysis, namely sulfuric, hydrochloric, phosphoric, hydrobromic, and nitric acids [48]. Sulfuric acid is the most commonly used acid because it can introduce negatively charged sulfate groups on the surface of cellulose fibers, which leads to the high colloidal stability of the resultant CNC suspension [49]. Hydrochloric acid, otherwise, tends to flocculate the CNC particles, making a less stable aqueous suspension due to the absence of charged groups on the CNC surface [50]. The agglomeration of the crystals is prevented by the repulsion of surface groups with the same charge [51].

The acid hydrolysis method is widely used because it is relatively simple and scalable and allows for the formation of nanoparticles with high crystallinity, great surface reactivity (high density of hydroxyl groups on the surface), and high aspect ratio (elongated rod-like particles). The main disadvantage is the use of strong acids, which can lead to the degradation of the nanoparticles and raise environmental concerns.

#### 3.1.2. Enzymatic Hydrolysis

Enzymatic hydrolysis is also used to obtain CNCs by a process that involves cellulolytic enzymes to cleave the cellulose fibers into nanocrystals. Cellulolytic enzymes (mainly cellulases) catalyze the hydrolysis of the glycosidic bonds between the glucose units. The mixture obtained is subjected to centrifugation to separate the nanocrystals from the remaining cellulose fibers and enzyme solution.

The glycosidic bonds of cellulose fibers can be the subject of selective enzymatic hydrolysis, depending on the cellulolytic enzymes used. Endoglucanases and cellobiohydrolases will preferentially attack the amorphous and crystalline regions of cellulose fibers, respectively [52].

The enzymatic hydrolysis method presents several advantages, especially in terms of environmental problems (it does not involve the use of strong acids or other hazardous chemicals), milder conditions (which can help preserve the structural integrity of the nanoparticles), and reduced defects of the obtained CNCs (which lead to nanoparticles with a more uniform size distribution). The principal challenges using this method are the enzyme’s activity and stability, which depend on the temperature, incubation time, pH, and inhibitors; the slower reaction rates due to the inherent nature of enzymatic reactions; and the cost of cellulolytic enzymes.

### 3.2. Cellulose Nanofibrils

CNFs are typically produced through mechanical treatment, which involves the disintegration of the cellulose fibers, preserving some of the amorphous regions, and thus contributing to the formation of a network structure with nano-sized fibers and a high specific surface area. Other interesting CNF characteristics are easy surface modification and functionalization, excellent mechanical properties, and good thermal stability [34,51]. CNFs are typically composed of long and flexible entangled particles with dimensions of up to more than 1000 nm in length and 5–100 nm in diameter. 

#### 3.2.1. Mechanical Methods

Mechanical methods such as high-pressure homogenization, microfluidization, grinding, and high-intensity ultrasonication are among the most common methods used to produce CNFs from cellulose fibers. The main achievement of mechanical methods is the non-use of organic solvents, which is very attractive in terms of environmental impacts [53]. Mechanical methods have, however, a great challenge to overcome related to high energy consumption [8].

High-pressure homogenization involves forcing a cellulose fiber–water suspension through a narrow gap at high pressures. The shear forces generated by the rapid flow through the gap cause cellulose fibers to break down into nanofibrils. The fibrillation degree will depend on the number of times the suspension passes through the homogenizer and the applied pressure [54]. High-pressure homogenization has an important limitation, which is the clogging of the system when using long fibers [55,56].

Microfluidization uses an intensifier pump to enhance pressure and is similar to high-pressure homogenization. The cellulose fiber–water suspension is passed through a thin chamber with a specific geometry at high pressures and high velocities. Intense turbulence flow and shear forces are achieved, leading to cellulose fibrillation. The number of passes through the microfluidizer and the chamber sizes will define the extent of fibrillation [57]. Microfluidization has the same drawback as high-pressure homogenization, that is, the clogging situation [56].

Grinding has been used for centuries in the papermaking industry and must be used carefully to ensure that it does not cause excessive damage to the nanofibrils, resulting in poor physical strength, crystallinity, and thermal stability [58,59]. The cellulose fiber pulp is passed through a couple of stones, where one is fixed and the other is rotating. Shear forces created by mechanical stress are responsible for the cellulose fibers breaking down into nanofibrils. The fibrillation degree will depend on the distance between the stones and the number of pulp passes [54]. The clogging of the system found in high-pressure homogenization and microfluidization can be minimized by adjusting the distance between the stones [38].

High-intensity ultrasonication involves subjecting a cellulose fiber–liquid suspension to high-intensity ultrasonic waves (>20 kHz). Tiny bubbles in the liquid are formed, which rapidly grow and implode, generating intense shear forces that lead to cellulose fibrillation. Cellulose fiber suspension consistency, sonication time, temperature, and power will define the extent of fibrillation [37,60].

#### 3.2.2. Combination of Methods

A combination of mechanical methods with enzymatic or chemical pre-treatments can be an alternative to obtaining CNFs from cellulose fibers with lower energy consumption. The enzymatic pre-treatments break down the glycosidic bonds in cellulose chains, helping to improve the formation of nanofibrils, and the chemical pre-treatments introduce charged functional groups (positive or negative) on the cellulose fiber surface, generating repulsive forces and improving the fibrillation degree. Repulsive forces are responsible for turning the hydrogen bonds in cellulose fibers weaker, making it easier to separate the nanofibrils (fibrillation). The most common pre-treatments are enzymatic hydrolysis, 2,2,6,6-tetramethylpiperidine-1-oxyl (TEMPO)-mediated oxidation, carboxymethylation, cationization, and ionic liquid/deep eutectic solvent treatments.

Enzymatic hydrolysis is a pre-treatment similar to the method used to produce CNCs and involves the use of enzymes (specifically cellulases). The cellulases are applied to cellulose fiber suspensions and initiate the hydrolysis of the amorphous regions of the fibers, increasing their accessibility to mechanical cellulose fibrillation. Lignin must be removed prior to enzymatic hydrolysis due to its recalcitrance and interaction with cellulases [61]. The non-toxic nature of the enzymes and the non-production of hazardous by-products make enzymatic hydrolysis a more environmentally friendly pre-treatment compared to chemical pre-treatments [62].

TEMPO-mediated oxidation is a well-known chemical pre-treatment and one of the most common methods for cellulose fiber surface modification. The cellulose fibers are soaked in a solution containing TEMPO (catalyst), NaClO (oxidant), and NaBr under alkaline conditions. Negative charges are introduced on the cellulose fiber surface through the selective oxidation of primary hydroxyl groups to carboxylate groups (COO^−^) via aldehyde groups, leading to repulsion forces that facilitate the mechanical cellulose fibrillation. The formation of carboxylate groups is directly proportional to the amount of NaClO used and also to the oxidation time up to a certain degree of substitution [63].

Carboxymethylation is an alternative chemical pre-treatment to TEMPO-mediated oxidation. The process involves the treatment of cellulose fibers with an alkali solution (typically NaOH) to activate the hydroxyl groups, which will react with monochloroacetic acid or sodium chloroacetate. Carboxymethyl groups (CH_2_COO^−^) are introduced on the cellulose fiber surface, generating repulsion forces that enhance the mechanical cellulose fibrillation.

Cationization is a common chemical pre-treatment and another method for cellulose fiber surface modification. The hydroxyl groups, after activation through the treatment of cellulose fibers with an alkali solution (NaOH), react with cationic reagents such as 2,3-epoxypropyl trimethylammonium chloride (EPTMAC) and chlorocholine chloride. The cationic groups introduced on the cellulose fiber surface promote the mechanical cellulose fibrillation.

### 3.3. Bacterial Nanocellulose

BNC is obtained from certain bacteria species, including *Gluconacetobacter*, *Agrobacterium*, *Rhizobium*, *Salmonella*, *Escherichia*, and *Sarcina*, with *Gluconacetobacter* being the most commonly used [64,65]. The bacteria ferment low molecular-weight sugars to synthesize a well-organized, dense, and coherent structural network of nanocellulose. BNC is exceptionally pure and has a high degree of crystallinity because it is composed of pure cellulose with minimal impurities or amorphous regions. The main properties of BNC are their excellent mechanical strength, good thermal stability, excellent water retention, and biocompatibility [66]. In terms of morphology, BNC consists of twisted ribbon particles with dimensions of 1000–5000 nm in length and less than 100 nm in diameter. The most common sources are low molecular-weight sugars such as glucose, fructose, sucrose, arabitol, and mannitol [35].

Bacterial fermentation is a process where bacteria are inoculated in an aqueous medium to grow through consumption of the carbon source (often glucose) and other essential nutrients such as nitrogen sources and mineral salts. The synthesis of nanocellulose occurs at the same time during the bacterial fermentation by self-assembling in a pellicle at the air–liquid interface of the culture medium. The pellicle is composed of a complex, entangled network and can be carefully harvested through gentle lifting or filtration. The harvested pellicle is washed to remove any remaining bacterial cells, nutrients, and by-products. Nanocelluloses may then be dried (air drying or freeze drying) to remove excess water and make them easier to handle.

The synthesis of BNC is relatively expensive, given the use of specialized fermentation equipment and long production times (ranging from a few days up to two weeks) [37,67].

### 3.4. Properties and Applications

Nanocelluloses exhibit remarkable properties, including biodegradability (they can replace non-biodegradable materials), renewability (produced from cellulose, which is abundant in nature), biocompatibility (they can be used in a biological system without causing any harm to it), high aspect ratio and surface area (which may improve interactions with other materials), transparency (affording transparent films and suspensions), surface functionalization ability (they can be modified by the incorporation of new functional groups), barrier properties (which enhance the shield against gases and liquids), hydrophilicity (improving the dispersion in water-based systems), excellent mechanical properties (high tensile strength and Young’s modulus), and tunable electrical conductivity (allowing the creation of materials from insulating to conductive) [36,68,69].

Nanocelluloses are very versatile and valuable materials with a wide range of applications, such as in electronic devices (transparent and flexible films), construction (reinforcing agents), automotive industry (lightweight and sustainable materials), biomedical and pharmaceutical areas (drug delivery systems), cosmetics (texture and stability of lotions), packaging (biodegradable materials), coatings (water repellency), textiles for clothing (breathable and moisture-wicking sportswear), and environmental (water purification processes) [3,36,44,70]. An emerging application is in the preservation of cultural heritage, as nanocelluloses possess excellent optical and mechanical properties crucial to maintaining the visual integrity of documents and making the documents more resistant to tearing and degradation over time.

## 4. Conservation and Restoration of Historical Documents

Paper was originally produced from rags of linen, hemp, and cotton [71], which consisted of crystalline cellulose of high quality with strong structure and excellent chemical stability. The demand for paper in the mid-nineteenth century, the invention of the printing press, and the scarcity of rag sources made it necessary to search for other raw materials. The paper has played a crucial role for centuries and continues to be an important medium for recording and communicating information in the form of books, scientific journals, magazines, newspapers, historical documents, and archives. The deterioration of paper is still a problem, caused by many factors such as acid and alkaline hydrolysis, oxidation processes, inks and pigments, insects and microorganisms, humidity and light absorption, air pollution, or storage conditions [72]. Mechanical damages like tears, cuts, and deformations can also deteriorate paper. The most common degradation process is acid hydrolysis, which contributes to a significant decrease in the mechanical properties of cellulose [73,74,75]. Acid hydrolysis can be caused by acids released from the degradation of cellulose and hemicelluloses with aging (formic, lactic, and oxalic acids), by the presence of non-removed lignin, by the absorption of air pollutants (mostly sulfur and nitrogen oxides), and by the inks used in writing [76].

The conservation and restoration of cultural heritage is a promising and innovative application for nanocelluloses. Nanocelluloses can be used as (i) a consolidating agent to prevent further deterioration of fragile materials; (ii) an adhesive to repair and restore damaged or detached components; (iii) an ink and pigment stabilizer to prevent bleeding or fading of inks and pigments; (iv) an anti-aging agent by creating a protective barrier against oxidative and environmental damages; (v) a humidity controller to maintain stable environmental conditions; (vi) an ultraviolet radiation protector to prevent fading and deterioration of the artifacts and artworks; and (vii) an antibacterial agent by inhibiting the growth of microorganisms.

### 4.1. Iron Gall Ink

Documents on paper written with iron gall ink often require careful conservation and restoration efforts. The acidity of the iron gall ink can lead to the degradation of paper, including weakening, embrittlement, and discoloration, resulting in so-called iron gall ink corrosion. Conservators use specialized techniques to stabilize and repair documents that may have been damaged by the iron gall ink.

Iron gall ink was widely used for centuries, from the Roman Empire to the Renaissance, and is the primary ink used for writing and drawing in Europe and the Middle East during the medieval and early modern periods. The documents written with iron gall ink present a blue–black color. However, over time, the iron gall ink can darken and turn blacker due to the oxidation of iron ions. Iron gall ink was commonly used for legal documents, manuscripts, religious texts, letters, and other important written records due to its dark color, resistance to fading, and excellent adhesion to paper. The permanent character of iron gall ink on the respective cellulosic support provided high protection against the forgery of documents, contributing to its popularity [77].

The formation of iron gall ink results from the combination of iron salt (ferrous sulfate, also called green vitriol), tannic acids often derived from galls, bark, leaves, roots, or other plant materials (gallic acid), a binding agent (gum Arabic), and a solvent (water, beer, or wine) (Figure 2). The tannin-rich sources are crushed and soaked in the solvent to extract gallic acid that will react with ferrous (II) sulfate to produce a blue–black iron (III)-tannin complex, known as ferrogallotannate or ferrotannate, which is insoluble in water. Gum Arabic, soluble in water, is added to increase the viscosity, to keep the pigment particles in suspension, and to bind the ink to the writing surface. The mechanism consists of the formation of the iron (III)-tannin complex from the air oxidation of the iron (II)-tannin complex, which is colorless and soluble in water. A low pH of iron gall ink between 2 and 3 confers a high acidity to the paper, requiring proper storage in controlled environments (light, temperature, and humidity) [78].

Iron gall ink was replaced by modern commercial ink because it is corrosive and can damage paper, putting at risk a significant part of the cultural heritage. For instance, the complete works of Victor Hugo almost disappeared, and 60–70% of Leonardo da Vinci’s legacy presents signs of deterioration [79]. The main reasons for the corrosion caused by iron gall ink are acid hydrolysis and oxidation of cellulose, which contribute to the weakening and discoloration [80,81,82]. Acid hydrolysis results from the acidity of the iron gall ink components and the sulfuric acid produced during the preparation of the iron gall ink, where both act as catalysts leading to the chain scission of cellulose. The process can continue over the centuries, except if the acid is neutralized, contributing to a loss of the paper’s mechanical strength [83]. On the other hand, the cellulose oxidation is caused by the excess of iron (II) ions used to produce the iron gall ink, which catalyzes the formation of very reactive hydroxyl radicals from hydrogen peroxide by the Fenton reaction [84]. Hydrogen peroxide is formed during the reduction of atmospheric oxygen by iron (II) ions. Hydroxyl radicals start a sequence of radical reactions resulting in the scission and cross-linking of cellulose, thus contributing to a loss of mechanical strength and a decrease in the water absorption of the paper structure [85]. The localization of the paper degradation can surround the ink through the long migration of sulfuric acid in the acid hydrolysis or can be close to the ink through the short migration of iron (II) ions in the oxidation process [85].

### 4.2. Conservation and Restoration Methods

Conservation and restoration methods for documents on paper are essential to preserving and protecting cultural heritage. The main goals are to prevent degradation, repair the damage, improve the legibility, maintain integrity, and extend the lifetime of the documents. Conservation and restoration methods involve several main steps, namely: (i) assessment and documentation to identify physical damage or deterioration; (ii) surface cleaning to remove dirt, dust, and surface contaminants; (iii) testing and analysis to determine the paper composition, ink type, and chemical issues in order to choose the best method to apply; (iv) deacidification to neutralize acidic compounds that can lead to degradation; (v) consolidation and mending using an appropriate method to reinforce weakened areas and treat tears and losses (including the use of Japanese Paper and the calcium phytate method); (vi) final checks to ensure that the work has been successfully completed and the document is stable, preserved, and ready for storage or display. Some of the most common methods used in the conservation and restoration of paper documents are described below.

#### 4.2.1. Surface Cleaning

Surface cleaning can include disinfestation (elimination of rodents and insects) and disinfection/sterilization (extermination of bacteria, viruses, and fungi) to protect the artwork from biological agents [86]. The most common methods for disinfestation are (i) fumigation with chemicals like methyl bromide or sulfuryl fluoride to kill or repel pests; (ii) anoxia, involving the creation of an oxygen-free environment to suffocate pests; and (iii) cold storage, consisting of subjecting the paper to very low temperatures, slowing down or stopping the activity of pests. The disinfection/sterilization processes include (i) ultraviolet light treatment consisting of the exposition of paper to ultraviolet light to kill surface microorganisms [87]; (ii) autoclaving by subjecting the paper to high-temperature steam under pressure to kill all microorganisms; and (iii) gamma radiation treatment involving the irradiation of paper with gamma radiation to sterilize paper [88].

Mechanical cleaning is another method of surface cleaning used to remove dust, dirt, and contaminants from the paper surface for aesthetic reasons [86]. The most typical techniques include (i) soft brushes made of natural hair or synthetic materials to gently brush away dust and dirt; (ii) special sponges designed to attract and absorb dirt and contaminants; and (iii) rubber erasers, such as vinyl or gum erasers, used for gentle rubbing to lift off surface dirt. Mechanical cleaning relies on the application of mechanical energy without the use of solvents or liquids. However, this method can be a disruptive process that may cause abrasion of the paper surface, forcing contaminants into the paper structure and causing severe damage.

#### 4.2.2. Deacidification

Deacidification of paper is a chemical process that involves the introduction of alkaline substances to neutralize the acids and to create an alkaline reserve to neutralize any acidic substance that might be formed [89,90,91]. High acidity levels contribute to the deterioration of paper, causing brittleness and instability. Deacidification helps to stabilize, prevent further deterioration, and prolong the lifetime of paper documents [72]; however, it cannot restore the loss of mechanical strength [86]. Liquid (aqueous and organic solvents) and gaseous processes can be used. Aqueous deacidification involves the use of a liquid solution containing alkaline agents (calcium bicarbonate or magnesium bicarbonate) where the paper is immersed or sprayed. This process causes fiber swelling and color bleeding and cannot be used on documents that are sensitive to water or contain water-soluble inks. Organic solvent deacidification uses volatile organic solvents with dissolved alkaline agents (magnesium carbonate in methanol or barium hydroxide in methanol), and the paper is soaked or sprayed. This process wets the paper more rapidly, provokes less fiber swelling, and can be employed on books and other bound documents. The main drawback is the flammability and toxicity of the organic solvents. Gaseous deacidification involves exposing the paper to alkaline gases (ammonia or amines) within a sealed chamber or enclosure. This process may not produce the amount of alkaline reserve necessary, and some deacidification agents may introduce the risk of burning and explosion. The principal advantage is its suitability to be employed on books and other bound documents. The documents can be treated individually as single paper sheets (aqueous deacidification) or in groups as books and bound documents (organic solvent deacidification and gaseous deacidification). A mass deacidification process has been developed to rescue books and bound documents, which is very useful for libraries, archives, and institutions with extensive collections [92,93].

#### 4.2.3. Japanese Paper

Japanese Paper, often referred to as washi, is a traditional handmade paper that has been used for centuries in Japan and is among the best materials for conservation and restoration methods [94]. Japanese Paper can be obtained from three different types of fibers: kozo bush (the longest fiber that makes the strongest paper), mitsumata shrub (softer and shorter fiber with a warm tone), or gampi tree (noble fiber with an exquisite natural sheen). The most commonly used material is kozo, which can be found in 90% of all Japanese Paper used for conservation and restoration purposes [94]. The kozo fibers are cooked with a mild alkaline solution of wood ash (K_2_CO_3_), soda ash (Na_2_CO_3_), or lime (Ca(OH)_2_) in traditional methods or a strong alkaline solution of caustic soda (NaOH) in modern methods to help break down the fibers and to remove lignin and other impurities. Japanese Paper presents unique properties, namely: (i) longevity (is made from fibers that are naturally resistant to deterioration); (ii) acid-free (prevents acid migration and degradation of the repaired area over time); (iii) strength and durability (reinforces weakened or torn areas); (iv) semi-transparency (allows to see and/or read the printed or handwritten text); (v) flexibility (adapts to the contours of the repaired area); (vi) adhesion ability (creates strong bonds that are reversible and can be easily removed if necessary); and (vii) reversibility in the adhesion (allows to remove or adjust repairs without causing harm to the original document). The basis for mending a document with this material is to apply a thin layer of adhesive to Japanese Paper or to the original document, place one on top of the other, and press down gently to ensure that they are glued together. The paper surface must be cleaned and free from dust, dirt, and contaminants before Japanese Paper is applied. The used adhesives need to have enough bonding strength, resistance to aging, color stability, and chemical inactivity, and be easily removed without using severe conditions, toxic solvents, or complicated procedures [86]. Wheat starch paste is the most common adhesive and can be simply removed by adding moisture. Other adhesives are gelatin, chitosan, hydroxypropyl cellulose, methylcellulose, polyvinyl alcohol, and polyethylene [86,95]. Japanese Paper is available in a wide range of types and thicknesses that can be used in machine-made papers, handmade papers, brittle papers, manuscripts, and drawings, where the focus is to stabilize and reinforce the damaged paper-based materials.

#### 4.2.4. Calcium Phytate Method

The calcium phytate method is, presently, the most effective method used for the conservation and restoration of paper documents written with iron gall ink. It was proposed for the first time by Neevel in 1999. Phytic acid is a natural antioxidant present in the seeds of plants that can block the oxidation process of unsaturated fatty acids catalyzed by iron. Phytate is the anion of the phytic acid, myo-inositol hexakisphosphate or inositol polyphosphate, a strong chelating or complexing agent of important metal ions. The “free” iron ions (not coordinated with tannins) responsible for the cellulose oxidation process can be inactivated by phytates [96]. The reaction between the iron ions and phytates leads to the formation of high-affinity complexes, preventing the participation of iron ions in Fenton oxidation reactions [97]. Various salts of phytic acid, such as sodium, magnesium, and calcium salts, were tested, and all were equally effective with some side effects [84]. The sodium phytate is very soluble and can migrate to the borders of the paper during drying, causing brownish discoloration. The treatment with magnesium phytate tends to produce brown ink and yellow paper. The calcium phytate allows the migration of the ink, changing the color. Treatments with phytates must always be followed by deacidification to neutralize the acidity of paper and prevent cellulose acid hydrolysis [86]. The calcium phytate method involves the immersion of paper in a calcium phytate solution with a pH between 5.0 and 6.0 (adjusted with ammonia), where the calcium ions can exchange with the iron (II) and iron (III) ions, forming iron–phytate complexes. Paper is also immersed in a gelatin solution bath to add a protective film between the atmosphere and the surface of the ink, increasing the mechanical strength and flexibility of paper [98]. The gelatin is used as a resizing agent for paper documents written with iron gall ink since it possesses a better blocking effect against ink corrosion compared to other adhesives commonly used for paper conservation [99]. The calcium phytate method was developed specifically for paper documents written with iron gall ink, and several studies have proven that it is very effective [100,101].

## 5. Nanocelluloses in Historical Documents

One of the methods used by conservators to restore damaged paper is the application of a reinforcing layer on the paper surface, known as lining, which allows the loss of mechanical resistance to be recovered [102]. The material typically used for this purpose is Japanese Paper, which provides high resistance, but it is not entirely satisfactory for all documents because the sheets are not very homogeneous [103]. Nanocelluloses have emerged as a novel and innovative approach to the conservation and restoration of paper documents because they can solve many challenges, such as structural weaknesses, pH instability, moisture exposure, and damage repair. The natural compatibility with the paper matrix (cellulose) and the inherent characteristics of nanocelluloses allow them to form strong, transparent, and fibrillar networks, which is very important for paper stabilization [74]. Nanocelluloses act as a coating layer for protecting diverse cellulose-based materials, leading to consolidation, strengthening, and improvement of barrier properties [68,104,105]. The protective coating should not change the physical, chemical, or optical properties of the paper substrate, including shine or color saturation [106]. Nanocelluloses are adequate because they are transparent, act as a nano-lining, and tend to accumulate on the surface of the paper with limited penetration [74,107,108,109]. The use of nanocelluloses also meets the requirement for sustainable materials based on renewable and biodegradable resources [110].

The three types of nanocelluloses (CNCs, CNFs, and BNC) have been employed in several studies in the field of conservation and restoration of paper documents, which will be reported in chronological order.

Santos et al. [111] compared the application of BNC with Japanese Paper to reinforce books from 1940 to 1960. Similar mechanical properties were found with an improvement in the optical properties (better legibility of letters) when using BNC compared to Japanese Paper (Figure 3). Paper reinforced with BNC exhibited Gurley air resistance higher than 900 s/100 mL, while Japanese Paper showed values between 20 and 40 s/100 mL. BNC provided high stability and enhanced the quality of deteriorated paper over time. The results indicated that the closed structure of BNC offers adequate protection against humidity and atmospheric pollutants. 

Völkel et al. [74] tested BNC suspensions and a suspension of CNF obtained from wood pulp by only mechanical homogenization to stabilize rag papers from the seventeenth to the nineteenth century and book papers from the twentieth century. The mechanically damaged areas were treated, and the weakened areas were consolidated without additional adhesive and negative side effects in the long term (Figure 4). 

Jia et al. [112] prepared a coating based on CNC and zinc oxide suspensions using Klucel as the adhesive to protect a school newspaper dating back to 1960 belonging to the Renmin University of China. An improvement in mechanical properties (tensile strength from 2.7 to 3.2 kNm^−1^), higher antibacterial and antifungal activity (*Aspergillus niger* colonial area of 75.6% and 7.8%), and higher protection against UV light (color difference of 3.7 and 1.6) were revealed for treated paper in comparison with untreated paper. The microbiological tests were made with two bacteria found in everyday life (*Staphylococcus aureus* and *Escherichia coli*) and five fungi often detected in archives or museums (*Aspergillus niger*, *Aspergillus versicolor*, *Rhizopus nigricans*, *Saccharomycetes*, and *Mucor*).

Camargos et al. [113] studied an aqueous dispersion of CNCs, calcium carbonate, propylene glycol, and methylcellulose as an alternative material to fill lacunae of paper sheets from a book printed in the twentieth century. Calcium carbonate was the filler, propylene glycol was the plasticizer, and methylcellulose was the sizing agent. The properties of the CNC-based composite were compared with the properties of conventional paper. The crystallinity index of the CNC-based composite (80.8%) was around three times higher than that of the conventional paper (27.2%). The CNC-based composite presented a mild basic character (pH of 7.5), which is advantageous over the acidity of the conventional paper (pH of 6.0). Similar mechanical properties were found in the CNC-based composite (Young’s modulus of 310 MPa) and conventional paper (Young’s modulus of 306 MPa).

Dreyfuss-Deseigne [114,115] developed CNF films to mend paper viewing slides from the mid-nineteenth century belonging to the French Museum of Cinema. The remarkable transparency and the very good stability to light, temperature, and humidity aging allowed the treatment of structural problems in the slides, such as weaknesses, tears, and losses (Figure 5). The CNF film and different types of Japanese Paper (gampi paper, hemp paper, kozo paper, and tengujo kozo paper) were compared for a few physical properties [114]. Among the Japanese Papers, the gampi paper and the tengujo kozo paper showed the best performance. However, almost all the results were worse than those obtained for the CNF film, such as luminance (40% for tengujo kozo paper and 38% for CNF film), color changes during light aging (0.8 for tengujo kozo paper and 0.3 for CNF film), and thickness (27 μm for gampi paper and 11 μm for CNF film). The CNF film and the Japanese Papers were also combined with four adhesives generally utilized in paper conservation (wheat starch paste, methylhydroxyethylcellulose, hydroxypropylcellulose (Klucel G), and Culminal MC2000). The Klucel G provided the best results compared to the other types of adhesives in terms of tension, planar distortion, and adhesion. This adhesive was the only one that could be directly used with the CNF film because it is ethanol-based and does not cause shrinkage, unlike the others, which are water-based adhesives. 

Xu et al. [106] studied a self-healing coating based on a composite of modified nanocellulose, calcium carbonate, fluorineacrylamide styrene, and acrylic copolymer to protect paper cultural relics. The calcium carbonate microcapsules were claimed to be responsible for the properties of self-healing and could release the inside healing agents to repair the film cracks when necessary. A comparison between the composite without and with calcium carbonate was made, demonstrating an increase in hardness (from 87.7 to 94.3 shore A), tensile strength (from 2.3 to 5.4 MPa), and water resistance (contact angles of 68 and 98°). Another comparison between untreated paper relics and relics treated with the composite containing calcium carbonate was conducted, revealing an increase in thickness (from 73 to 82 μm), tensile strength (from 358 to 2068 Nm^−1^), and antiaging properties (mass loss rate of 13.3 and 2.4%).

Völkel et al. [116] tested a calcium phytate/calcium hydrogen carbonate treatment combined with the application of CNF suspensions to stabilize chemically and mechanically rag papers written with iron gall ink from a collection of sermons from 1839 and 1840. The viability of the procedure was confirmed with the benefits of chemical stabilization (deacidification) of iron gall ink with calcium phytate/calcium hydrogen carbonate treatment and an additional significant mechanical stabilization of iron gall ink paper with the CNF application. Damaged samples, which are very sensitive to being handled with the risk of material loss, could be chemically and mechanically stabilized. Fractures, cracks, and imperfections were closed and sealed, making the samples manageable with a minimum risk of future damage (Figure 6). CNFs acted as a protective layer and had a minimum influence on the optical and haptic properties of the manuscripts. The integration of CNFs into the calcium phytate/calcium hydrogen carbonate treatment is important because an additional stabilization and drying step is avoided.

Operamolla et al. [117] employed sulfated and neutral CNC suspensions to restore and reinforce a book entitled “Breviarium Romanum ad usum Fratrum Minorum” from the eighteenth century. The paper samples treated with neutral CNC exhibited an improvement in optical quality and mechanical properties, such as maximum stress (7.8 MPa for untreated paper, 7.3 MPa for paper treated with sulfated CNC, and 12 MPa for paper treated with neutral CNC) and maximum strain (1.4% for untreated paper, 1.6% for paper treated with sulfated CNC, and 2.2% for paper treated with neutral CNC). This study also showed that the presence of surface sulfation may have a negative influence on the conservation of a paper artifact, compromising its pH and mechanical properties with aging.

Ma et al. [118] applied suspensions of CNCs with polyhexamethylene guanidine to reinforce ancient books. An excellent biocidal activity against *Aspergillus niger* and mixed mold (*Aspergillus niger*, *Trichoderma viride*, *Penicillium funiculosum*, and *Chaetomium globosum*) was demonstrated with a growth trace of less than 10% for treated paper (Figure 7). An improvement in mechanical properties (tearing strength of 3.6 and 4.7 mNm^2^g^−1^; tensile strain of 1.4 and 2.1%) and outstanding performance in aging tests were also observed for treated paper in comparison with untreated paper.

Henniges et al. [119] tested CNF films, prepared with and without methylcellulose as an “internal adhesive,” to repair historical papers from the collection of The National Archives of UK. The addition of methylcellulose to pure CNF films provided an increase in the transparency and mechanical strength of the films. The two CNF films were compared with two commercial nanocellulose films (Innovatech Nanopaper Art Paper and Innovatech Nanopaper Art Paper Pro) and one Japanese Paper (tengujo). The CNF films and the Innovatech Nanopaper Art Paper exhibited similar properties, unlike the Innovatech Nanopaper Art Paper Pro, possibly due to differences in the production process. Compared with the Japanese Paper, the CNF films showed an increased tendency to curl when moistened and lower permeability. The different films and Japanese Paper were combined with two adhesives (Klucel G and Isinglass) commonly utilized for mending tears by the conservators of The National Archives of UK. Klucel G used in tracing papers provided the best results for planar deformation but the worst results for adhesive strength compared with Isinglass. The latter adhesive generated the most significant distortion when applied to smooth films, such as Innovatech Nanopaper Art Paper Pro. Tracing papers repaired with CNF films containing methylcellulose presented a higher tensile strength than when repaired with the other types of films. This property was measured by the mass load required to break the tear mend, and the results with Klucel G were 1600 g for CNF films without methylcellulose, 1900 g for CNF films with methylcellulose, 1700 g for Innovatech Nanopaper Art Paper, 1200 g for Innovatech Nanopaper Art Paper Pro, and 700 g for Japanese Paper. The results with Isinglass were 800 g for CNF films without methylcellulose, 2200 g for CNF films with methylcellulose, 1300 g for Innovatech Nanopaper Art Paper, 1600 g for Innovatech Nanopaper Art Paper Pro, and 600 g for Japanese Paper. The best combination to repair historical tracing papers was the CNF film containing methylcellulose with Klucel G, affording satisfactory transparency, high strength after adhesion, stable aging properties, and potential for customization.

Völkel et al. [120] prepared suspensions of CNFs to stabilize historical papers damaged during the fire of the Duchess Anna Amalia Library in Germany in 2004. The artworks were from the eighteenth and early nineteenth centuries, made of rag paper, and a large part belonged to musical literature. The application of the CNF suspensions did not show a negative visual impact and provided mechanical stabilization in the long term, reducing paper fragility.

Camargos et al. [121] studied multifunctional coatings based on CNCs, CNFs, and lignin nanoparticles in aqueous dispersions to protect paper-based artifacts. The coatings were revealed to be effective against UV radiation and moist-heat aging (color difference of 6.5 for untreated paper and 1.9 for treated paper), protecting or attenuating the paper degradation. The surface morphology, roughness, and vapor permeability of the paper were mainly maintained after the coating application. A decrease in wettability (enhancement of water resistance) with a contact angle of <20° for untreated paper and 60° for treated paper was also observed (Figure 8).

Ma et al. [122] used superhydrophobic self-cleaning coatings based on CNCs, calcium carbonate, and polydimethylsiloxane suspensions to reinforce and re-repair historical books. The CNCs were responsible for the high mechanical strength achieved, and the calcium carbonate nanoparticles for the excellent properties of self-cleaning and deacidification (removal of the acidic substances). The coated historical paper was hydrophobically modified using methyltrimethoxysilane, which is a modifier of low surface energy. An increase in thickness (75 to 92 μm), water contact angle (87.5 to 152.5°), and roughness (3.8 to 4.3 μm) was demonstrated for treated paper. The self-cleaning property was confirmed by placing garden soil on the paper surface and then washing it. Untreated paper quickly became wet and contaminated, unlike treated paper, where water droplets rolled down from the paper surface (Figure 9). The pH value of treated paper (7.5–7.8) was appropriate for the purposes of paper preservation compared to untreated paper (5.3), since the recommended pH value is between 7.0 and 8.5. A general improvement in the mechanical properties was observed for treated paper in terms of tensile strength (28.6 to 33.9 Nmg^−1^) and tearing strength (3.8 to 4.3 mNm^2^g^−1^).

Elmetwaly et al. [123] applied multifunctional protective coatings based on inorganic nanotubes dispersed in CNCs to reinforce historical papers. An improvement in the tensile strength (from 12.4 to 15.8 N), elongation at break (from 0.3 to 0.8%), thermal stability, and UV protection (from 39.0 to 49.7) without altering the optical properties of the paper substrate was revealed.

All the studies showed that CNCs, CNFs, and BNC with or without other particles such as zinc oxide, calcium carbonate, polyhexamethylene guanidine, lignin nanoparticles, and inorganic nanotubes in suspensions or films are promising and innovative materials for reinforcing, mending, filling, protecting, and stabilizing historical documents. Table 2 presents a summary of the main results obtained from the applications of nanocelluloses in historical documents.

Each type of nanocellulose offers unique characteristics suitable for the conservation and restoration of historical paper documents. Generally, the different types of nanocelluloses exhibit exceptional mechanical properties (high stiffness and tensile strength), making them appropriate to reinforce fragile document substrates. However, due to the lower length of CNCs compared to CNFs and BNC and, consequently, lower nanofiber crosslinking, the mechanical properties are typically worse for CNCs. They can also add barrier properties against gases and liquids and enhance protection against environmental factors. CNFs and CNCs can be processed into thin and transparent films, providing protection without altering the appearance of the document. BNC is highly biocompatible, posing minimal risk to people. Its high purity is also an important feature, reducing their degradation over time. However, the production of BNC involves a time-consuming process and specialized equipment, leading to higher production costs. On this issue, CNFs can be produced using relatively simpler mechanical processes. The number of studies present in the literature is still limited to assess the relative efficiency of the different types of nanocelluloses for the purposes of conservation and restoration of historical documents. Studies comparing the different types of nanocelluloses are needed.

The most important application of nanocelluloses in the field of conservation and restoration of paper documents is as a reinforcing agent due to the formation of strong hydrogen bonds with the different hydroxyl groups among the cellulosic fibers. The inherent compatibility of nanocelluloses with paper results from the fact that cellulose is the primary component of paper, and nanocelluloses are derived from cellulose. Paper strength depends on paper fiber strength, the strength of the bond between fibers, and the tendency of entanglement of fibers in the paper [124]. The introduction of cellulose particles with nanoscale dimensions will contribute to filling the voids in the fiber-to-fiber contact area, increasing the number of hydrogen bonds formed, the bonded area, and thus improving the strength of paper without compromising the appearance or texture [125].

The use of Japanese Paper is the approach most employed in the conservation and restoration of paper documents. The long-standing traditional Japanese Paper offers a fibrous structure, strength, and flexibility to make it a reliable choice for repairing tears, reinforcing fragile areas, and infilling losses in historical documents. However, Japanese Paper has limitations regarding the fiber size and potential variability in quality, limited mechanical strength (especially at low grammages), low transparency, and, additionally, it visually structures the paper surfaces where is applied [74,114,126]. On the other hand, nanocelluloses are innovative materials with high transparency and remarkable mechanical properties, providing a significant reinforcing effect on paper and enhancing durability and document longevity. However, being a relatively recent alternative, the use of nanocelluloses requires in-depth study for long-term effects and presents some additional drawbacks related to methodology standardization, cost considerations, and large-scale production. Both approaches have advantages, and the choice depends on the specific conservation and restoration goals.

## 6. Conclusions

The most popular medium for recording and communicating information through the years has been paper, which can deteriorate with aging due to inks and pigments, insects and microorganisms, humidity and light absorption, air pollution, and storage conditions. Nanocelluloses are environmentally friendly and sustainable, representing a class of materials with extraordinary potential in the area of conservation and restoration of paper documents, allowing to improve the mechanical strength and maintain the authenticity and integrity of the artworks. Nanocelluloses can increase resistance to tears, wrinkles, and losses, which is particularly valuable for the handling and exhibition of historical documents. Their barrier properties against environmental factors slow down and/or inhibit degradation, enhancing the longevity of paper substrates. Nanocelluloses can be incorporated without causing harm to the artworks, and their application can be reversed, which is perfectly aligned with the principles of the conservation and restoration field. Additionally, it should be noted that the optical and mechanical properties of nanocelluloses are better compared with Japanese Paper giving more transparent and stronger films.

The efficacy of nanocelluloses in strengthening, reinforcing, and stabilizing historical documents could be particularly crucial in texts written with iron gall ink that have been weakened or compromised through the problem of iron gall ink corrosion. Nanocelluloses may prevent the migration and fading of iron gall ink, helping to increase the legibility and longevity of the artwork. Unfortunately, only one report was found on this matter. Systematic studies asserting the effects of nanocelluloses in the conservation and restoration of paper documents damaged by iron gall ink should be developed.

Overall, nanocelluloses represent a promising opportunity to revolutionize the conservation and restoration field, particularly concerning paper documents, offering versatile and effective techniques for safeguarding cultural heritage. In-depth research on nanocelluloses is required, involving interdisciplinary collaboration among scientists, conservators, and historians to solve some challenges such as easy availability, long-term stability, standardization of methodologies, cost-effectiveness, and ethical considerations. The availability of nanocelluloses needs to be increased by exploring diverse sources of cellulose and optimizing production methods. Long-term stability is critical and requires an understanding of the degradation mechanisms influenced by environmental conditions, chemical interactions, and aging processes. The standardization of methodologies needs to be well established for both production and application methods. Cost-effectiveness involves using renewable feedstocks, minimizing energy consumption, and assessing scalable production methods. The ethical considerations encompass the sustainability of raw materials, environmental impacts, responsible manufacturing practices, and social/cultural implications.

## Figures and Tables

**Figure 1 polymers-16-01227-f001:**
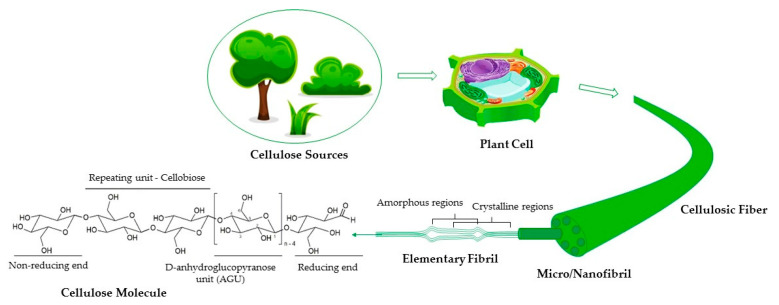
Hierarchical structure of cellulose (reproduced with permission from Almeida et al., 2023 [30], Copyright © 2023, Elsevier).

**Figure 2 polymers-16-01227-f002:**
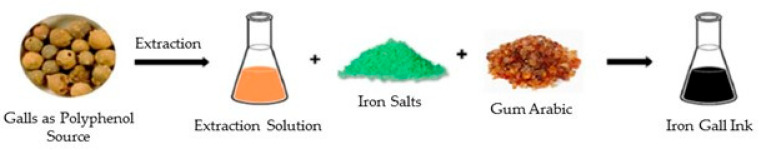
Main steps involved in the preparation of iron gall ink (reproduced with permission from Díaz Hidalgo et al. [18] Copyright © 2018, Springer).

**Figure 3 polymers-16-01227-f003:**
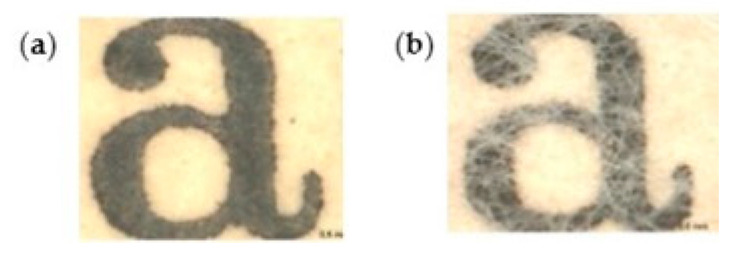
Microphotographs from a book from 1940 to 1960 lined with (**a**) bacterial cellulose and (**b**) Japanese Paper (reproduced with permission from Santos et al., [111] Copyright © 2016, Springer).

**Figure 4 polymers-16-01227-f004:**
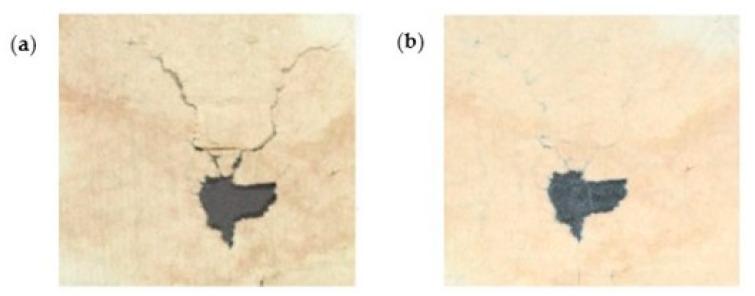
Mechanical damages such as cracks and losses: (**a**) before and (**b**) after a direct application of nanocellulose (reproduced with permission from Völkel et al., [74] Copyright © 2017, Springer).

**Figure 5 polymers-16-01227-f005:**
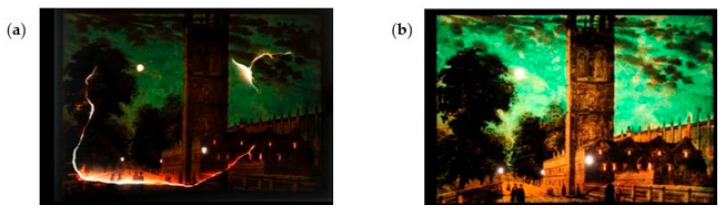
One viewing slide from the French Museum of Cinema under transmitted light: (**a**) before and (**b**) after treatment with microfibrillated cellulose film (reproduced with permission from Dreyfuss-Deseigne [114], Copyright © 2017, Taylor & Francis).

**Figure 6 polymers-16-01227-f006:**
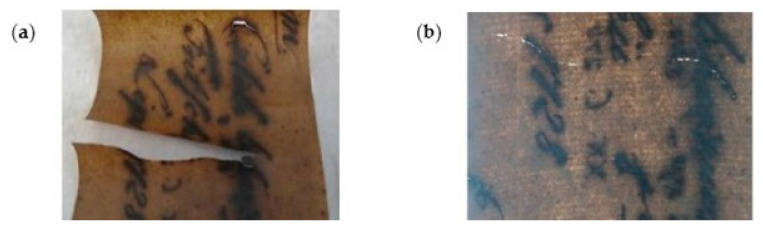
Damaged sample: (**a**) before and (**b**) after mechanical stabilization with nanofibrillated cellulose and chemical stabilization with calcium phytate/calcium hydrogen carbonate (reproduced with permission from Völkel et al., [116] Copyright © 2020, Springer).

**Figure 7 polymers-16-01227-f007:**
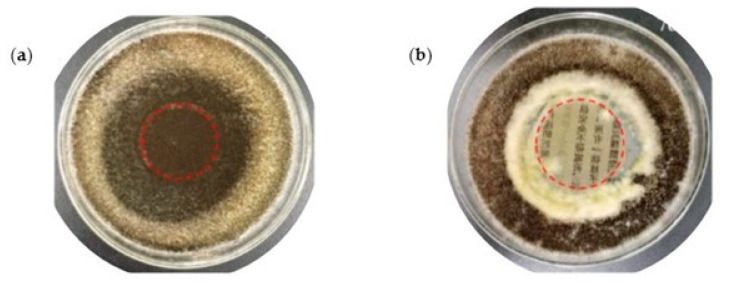
Effect of *Aspergillus niger* in ancient paper: (**a**) before and (**b**) after treatment with cellulose nanocrystals (reproduced with permission from Ma et al., [118] Copyright © 2021, Springer).

**Figure 8 polymers-16-01227-f008:**
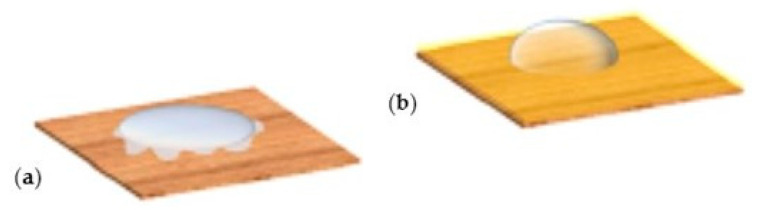
Effect of wettability: (**a**) before and (**b**) after treatment with a nanocomposite of cellulose nanocrystals, cellulose nanofibrils, and lignin nanoparticles (reproduced with permission from Camargos et al., [121] Copyright © 2022, American Chemical Society).

**Figure 9 polymers-16-01227-f009:**
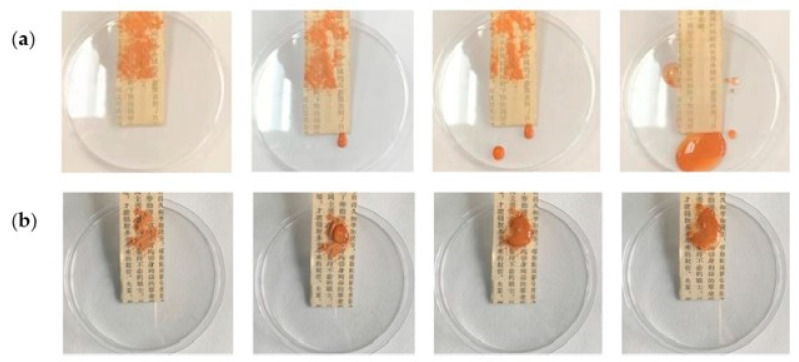
Effect of self-cleaning in historical paper: (**a**) before and (**b**) after treatment with a coating of cellulose nanocrystals, calcium carbonate, and polydimethylsiloxane (reproduced with permission from Ma et al., [122] Copyright © 2022, Elsevier).

**Table 1 polymers-16-01227-t001:** Comparison of different types of nanocelluloses.

Properties/Applications		Cellulose Nanocrystals	Cellulose Nanofibrils	Bacterial Nanocellulose
Morphology [40]		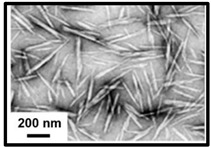	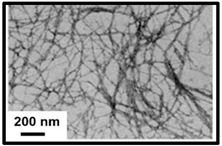	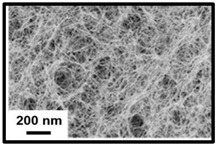
Preparation Process [39]		top–down	top–down	bottom–up
Size [3]	Length (nm)	50–500	>1000	1000–5000
	Diameter (nm)	3–50	5–100	<100
Crystallinity Index (%) [30}		54–88	<50	>88
Young’s Modulus (GPa) [41]		50–100	39–78	15–30
Purity [42]		low	low	high
Cost [3]		low	low	high
Main Applications [3]		optical devices, composite materials, and coatings	packaging, energy storage, and flexible electronics	antimicrobial products and flexible supercapacitors

**Table 2 polymers-16-01227-t002:** Principal results of nanocellulose applications are documented in historical documents.

Nanocellulose-Based Reinforcing Material	Nanocellulose Preparation Method	Document Type	Application Methodology	General Results	Reference
Bacterial nanocellulose (BNC)	BNC synthesized by *Gluconacetobacter sucrofermentans*	Book sheets from 1940 to 1960	Lining according to the traditional Japanese method (wheat starch as adhesive)	High stability over time; Burst and tear strength were improved; High legibility of the text; The air permeability of the lined BNC book sheets was reduced; Improvement of deteriorated paper quality	[111]
- Bacterial nanocellulose/carboxymethylcellulose - Cellulose nanofibrils (CNF)	- Commercial BNC/carboxymethylcellulose suspensions - Commercial mechanical CNF obtained from bleached softwood kraft pulp	- Rag papers from the 17th/18th/19th century - Newsprint paper from the 19th/20th century - Book papers from the 20th century	Suspensions were applied with a brush or film applicator on a vacuum panel + drying for at least 12 h	Low impact on the optical and haptic properties of the paper samples; BNC and CNF treatment did not exhibit any long-term negative side effects; The use of BNC and CNF enabled the consolidation of the mechanically damaged areas of paper	[74]
Cellulose nanocrystals (CNC)/zinc oxide	CNC prepared from microcrystalline cellulose by sulfuric acid hydrolysis	School newspaper from 1960	Suspensions applied by spraying and air-dried at room temperature (RT) (Klucel used as consolidating and dispersing agent)	Superior mechanical properties before and after aging; Antibacterial and antifungal activity; Increased resistance to dry-heat and UV light aging; Good color stability	[112]
Cellulose nanocrystals/propylene glycol/methylcellulose/calcium carbonate	CNC prepared from eucalyptus fibers by sulfuric acid hydrolysis	Two paper sheets from a book printed in the 20th century	The paper sheet holes were grafted with 3–5 layers of CNC-based composite suspension. Each layer was dried for 30–120 min under normal temperature and pressure conditions	The CNC-based composite graft exhibited a more regular and uniform surface compared to grafting with a suspension of cellulose fiber-based pulp	[113]
Cellulose nanofibrils	Commercial CNF obtained from birch kraft pulp	*Polyorama panoptique* viewing slides from the French Museum of Cinema	CNF film combined with 5% of Klucel G^®^ in ethanol (adhesive)	The CNF film with the 5% Klucel emerged as the better option for repairing the viewing slides, compared to four Japanese Papers and other adhesives; In terms of appearance, the tears in the viewing slides were effectively mended using the combination of CNF film and Klucel	[114]
Cellulose nanofibrils combined with calcium phytate/calcium hydrogen carbonate treatment	Commercial cellulose nanofibrils	Rag papers written with iron gall ink from a collection of sermons belonging to 1839 and 1840	CNF suspensions were applied on both sides of the paper with a brush on a vacuum panel	The effectiveness of the phytate treatment was not compromised by the incorporation of CNF; CNF had minimal influence on the optical and haptic properties of the manuscripts	[116]
Neutral and sulfated cellulose nanocrystals (N-CNCs and S-CNCs, respectively)	N-CNCs and S-CNCs prepared from Avicel^®^ by hydrochloric acid and sulfuric acid hydrolysis, respectively	Book pages from the 18th century	CNC suspensions were sonicated and then applied to the paper using a soft brush	The application of both types of CNCs did not affect the readability of the text on the book pages; N-CNCs treatment resulted in a stronger reinforcement of the paper compared to S-CNCs; The reversibility of the CNC coating was demonstrated	[117]
Cellulose nanocrystals/polyhexamethylene guanidine (PHMG)	CNC prepared from hardwood dissolving pulp by sulfuric acid hydrolysis	Paper samples from an old book published in 1954	The CNC/PHMG suspensions were sprayed twice on paper, air-dried at RT for 6–7 h, and then vacuum-dried in an oven (0.08 MPa, 50 °C) for 12 h	Improved mechanical properties (e.g., tearing and tensile strength, folding endurance) before and after aging; The presence of PHMG imparted strong antifungal activity to the treated paper	[118]
Cellulose nanofibrils/methycellulose (MC)	Commercial CNF obtained from bleached sulfite pulp from Norway Spruce	Tracing papers of a volume of registered designs (BT 43/58) in a collection of The National Archives of UK dating from 1859 to 1882	A strip of the CNF/MC film was single-sided brushed with 5% Klucel G^®^ in isopropanol (*w*/*v*, adhesive) and then applied under the tear in the tracing paper	The repair with CNF/MC film was barely noticeable after application; Satisfactory mechanical properties were attained for the historical tracing papers	[119]
Cellulose nanofibrils	Commercial CNF obtained from bleached sulfite pulp	Fire-damaged papers from the 18th and 19th centuries, originating from the Duchess Anna Amalia Library (Germany)	The CNF suspensions were airbrushed from the center to the edge of the paper. The treated paper was then dried at RT under 50% of relative humidity	The CNF treatment successfully mechanically stabilized the fragile and charred areas of paper; Minor influence on the visual appearance and legibility of the treated paper	[120]
Cellulose nanocrystals/calcium carbonate (CaCO3)/polydimethylsiloxane (PDMS) + Methyltrimethoxysilane (MTMS, hydrophobic modifier)	CNC prepared from hardwood dissolving pulp by sulfuric acid hydrolysis	Paper samples from an old book published in 1954	The CNC and CaCO3/PDMS suspensions were first sprayed onto paper. Then, the paper was hydrophobically modified by chemical vapor deposition using MTMS	CaCO3 in the CNC/CaCO3/PDMS treatment neutralized acids in historical paper and added an alkaline reserve; The hydrophobization of the paper-treated samples was confirmed by contact angle measurements; The presence of CNCs enhanced both the tensile and tearing strength of the treated papers before and after aging	[122]
Cellulose nanocrystals/halloysite nanotubes (HNTs)	CNC prepared from cotton fibers by sulfuric acid hydrolysis	Real written historical paper from a private collection dating back to 1943	The CNC/HNTs solutions were sprayed onto historical paper and then air- and oven-dried (90 °C; 30 min)	The CNC/HNTs coated layer had minimal impact on the optical properties of the historical paper and provided enhanced UV light protection; After coating, the tensile strength and elongation at the break of the historical paper were increased before and after thermal aging	[123]

## Data Availability

Not applicable.

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
