# Peer review of "Nanocelluloses and Their Applications in Conservation and Restoration of Historical Documents"

_polymers, 2024, doi:10.3390/polym16091227_

Round 1

Reviewer 1 Report

Comments and Suggestions for Authors

The image quality is low and needs improvement.

The type of nanocellulose should be precisely mentioned in the discussions (Section 5 of the MS and even elsewhere) from the three types of nanocellulose stated in the introduction.

It is unclear what is meant by nanocellulose in the discussions of the articles.

All names should be mentioned only once, not repeatedly, especially in subsequent instances. In subsequent instances, only the abbreviated name should be used. Use CNC, CNF, and BNC once and use these three in other places.

The explanation of bacterial nanocellulose is very general compared to the two previous types of cellulose and is very brief.

Sentences with similar meanings are repeated many times in different parts of the article, it is necessary to state them only once and delete them elsewhere.

The specifications of nanocellulose should be mentioned in Table 2.

A section titled "Advantages and Disadvantages of Using Three Types of Nanocellulose in Conservation and Restoration of Historical Documents" should be included in the article.

I suggest to include graphical image for conservation and restoration methods in text

The importance (advantages and disadvantages) of the three types of nanocellulose in conservation and restoration of historical documents for use should be mentioned in the abstract.

Comments on the Quality of English Language

need to be imporved

Author Response

Thank you very much for your comments and suggestions which have been carefully analysed. Most of them were taken into account.

  1. The image quality is low and needs improvement.

Response: The quality of some images was improved. The images were obtained from the available publications and reproduced with permission from the corresponding Editors. The quality of the reproduced images is limited by the quality of the original publications.

  1. The type of nanocellulose should be precisely mentioned in the discussions (Section 5 of the MS and even elsewhere) from the three types of nanocellulose stated in the introduction.

Response: As stated in the introduction, there are three types of nanocelluloses: cellulose nanofibrils (CNFs), cellulose nanocrystals (CNCs) and bacterial nanocelulose (BNC). The type of nanocellulose used in each study, in Section 5, has been specified according to these three types of nanocellulose.

  1. It is unclear what is meant by nanocellulose in the discussions of the articles.

Response: The response to this reviewer comment is related to the response to the previous comment. Nanocelluloses used in the studies described in section 5 (“Nanocelluloses in Historical Documents”) have been specified for each case. Note, as well, that in some cases, formulations involving not only nanocelluloses but composites with nanocelluloses (e.g. with mineral components or other polymers) have been employed for the purposes of conservation and restoration.

  1. All names should be mentioned only once, not repeatedly, especially in subsequent instances. In subsequent instances, only the abbreviated name should be used. Use CNC, CNF, and BNC once and use these three in other places.

Response: The manuscript was corrected according to this suggestion. The name of the three types of nanocellulose has been repeated only in figures and tables.

  1. The explanation of bacterial nanocellulose is very general compared to the two previous types of cellulose and is very brief.

Response: More information has been added in section 3.3, regarding Bacterial Nanocellulose (BNC). See lines 278-280 and 298-300. However, it should be noted that there is only one method of preparing BNC.  

  1. Sentences with similar meanings are repeated many times in different parts of the article, it is necessary to state them only once and delete them elsewhere.

Response: A thorough revision of the text was performed, deleting some of the repetitions.

  1. The specifications of nanocellulose should be mentioned in Table 2.

Response: A column with the specifications of the nanocelluloses used in the different studies has been added to Table 2 (see new column 2): the method of nanocellulose preparation was added.

  1. A section titled "Advantages and Disadvantages of Using Three Types of Nanocellulose in Conservation and Restoration of Historical Documents" should be included in the article.

Response: A paragraph has been added to section 5 (“Nanocelluloses in Historical Documents”) about the use of the three types of nanocelluloses in conservation and restoration of historical documents (lines 739-754).

  1. I suggest to include graphical image for conservation and restoration methods in text.

Response: This article is not focused on the description of the conservation and restoration methods. Therefore, we think that a graphical image is not necessary.

  1. The importance (advantages and disadvantages) of the three types of nanocellulose in conservation and restoration of historical documents for use should be mentioned in the abstract.

Response: The abstract was revised according to the reviewer suggestion.

Comments on the Quality of English Language

Need to be improved

Response: English Language has been revised.

Reviewer 2 Report

Comments and Suggestions for Authors

Well Written manuscript. Please address the suggestions.

Comments on the Quality of English Language

Good English. 

Author Response

Thank you very much for your comments and suggestions which have been carefully analysed and considered.

Well Written manuscript. Please address the suggestions.

The manuscript entitled “Nanocelluloses and their Applications in Conservation and Restoration of Historical Documents” by Marques et. al. is well written one and it may be accepted after minor revisions. The manuscript describes the importance of nanocellulose and their applications in conservation and restoration of historical documents. It has covered most of the publications in the above field.

I suggest few corrections for the betterment of the manuscript.

  1. Line 33: Please check the grammar. “Nanocomposites are a fibrous material”

Response: It was corrected to “Nanocomposites are fibrous materials”.

  1. Lines 114 &116: Please check the crystallinity index values for CNCs and CNFs

Response: We confirmed the crystallinity index values, which are correct according to the reference Fornari et al. (2022). The appropriate reference was now inserted.

iii. Line 333-336: Lacking clarity, please rephrase the sentence

Response: The sentence has been rephrased and we hope that it is now clearer (see new lines 338-341).

  1. Line 319: Authors may give a brief introduction to ‘Conservation and Restoration of Historical Documents’ before section 4, as there is lack of continuation from section 3 to section 4.

Response: A sentence has been added at the end of section 3.4 “Properties and Applications” to better connect the section 3 to section 4 (lines 320-323).

  1. Line 567-576: What the authors want convey or conclude from this paragraph. Please note.

Response: From this paragraph, it can be concluded that a coating based on cellulose nanocrystals and zinc oxide applied to an old document improved mechanical properties, also providing stronger antibacterial and antifungal activity, and greater protection against UV light to the treated document.   

  1. Figure 9b: With a contact angle 60 degrees, the picture doesn’t look like a hydrophilic one. Please check.

Response: We confirmed figure 9b. There is, in fact, a decrease in wettability of the treated paper-based artifacts. The contact angle of 60 degrees indicates moderate wettability, moderate water resistance and moderate hydrophilic character. Water contact angle gives an indication of the wettability of the solid. If it is above 90º, the solid is said to have poor wetting and is termed hydrophobic. If the contact angle is below 90º, the term hydrophilic is used.

Also, the authors may include more about the future prospects.

Response: “Conclusions” have been improved to include more information about the future prospects (lines 824-832).

Comments on the Quality of English Language

Good English

Round 2

Reviewer 1 Report

Comments and Suggestions for Authors

The revised manuscript is accepted